# Consequences of a Storm Surge for Aeolian Sand Transport on a Low-Gradient Beach

**Jorn T. Tuijnman †, Jasper J. A. Donker ‡ , Christian S. Schwarz § and Gerben Ruessink \***

Department of Physical Geography, Faculty of Geosciences, Utrecht University, P.O. Box 80.115, 3508 TC Utrecht, The Netherlands; jornt.tuijnman@gmail.com (J.T.T.); jasperdonker@gmail.com (J.J.A.D.); cschwarz@udel.edu (C.S.S.)

\*  Correspondence: b.g.ruessink@uu.nl; Tel.: +31-30-2532780

†  Current address: Aqua Vision, Servaasbolwerk 11, 3512 NK Utrecht, The Netherlands.

‡  Current address: Geodan, President Kennedylaan 1, 1079 MB Amsterdam, The Netherlands.

§  Current address: School of Marine Science and Policy, College of Earth, Ocean & Environment, University of Delaware, Newark, DE 19716, USA.

**Abstract:** Wind-blown beach sand is the primary source for the volume growth of the most seaward dune, the foredune. Strong wind events can potentially dominate long-term aeolian supply but in reality do not contribute considerably because they often coincide with a storm surge. The aim of this paper is to further our understanding of how a storm surge prevents or severely restricts aeolian supply. Using field data collected on the 1:50 sloping Egmond beach (Netherlands) in the aftermath of a 1-m storm surge, we show that the ground water in the upper beach rose to well above normal levels during the surge, which resulted in the development of a seepage face during falling tide and hence persistent saturation of the emerging beach. Using a fetch-based model, we predicted aeolian supply during the 2-day surge period to be about 66% of the potential supply. Fetch limitations imposed by the surge-induced inundation and the continuous saturation of the sand on the emerging beach both contributed to the predicted supply limitation. Our results quantitatively support earlier studies that suggested surges to be the primary condition that causes predictions of long-term potential foredune growth to overestimate measured growth.

**Keywords:** aeolian processes; surface moisture; storm surge; supply limitations; fetch

## 1. Introduction

Coastal foredunes are common morphological features along the world's sandy wave-dominated beaches and barrier systems [1,2]. Their dynamics reflect a sand-sharing system with the neighboring beach. During a single high-intensity storm or a cluster of storms a foredune can be eroded severely by marine processes [3–5], with the eroded sand being deposited on the beach and in the nearshore zone. The key hydrodynamic processes controlling dune erosion are reasonably well understood [4,6,7] and, accordingly, process-based models [8] can hindcast observed erosion events fairly well [5,9]. Foredune growth and recovery takes place through the deposition of wind-blown beach sand in vegetation and is a slow process in the sense that it may take seasons, years or longer before a previously eroded volume has been restored [10]. Accordingly, foredune growth is the cumulative effect of numerous aeolian transport events that differ in magnitude and duration [11]. A simple approach to compute the aeolian supply is the use of an equilibrium (or, potential) transport equation, in which the transport is a function of grain size and wind speed. Wind direction is additionally included to reflect that only onshore supply contributes to foredune growth [12,13]. Although the potential supply has been shown to match foredune volume growth reasonably well in some cases [14–16], it more often exceeds measured growth substantially [12,14,17–19].

Model failure may, at least in part, be due to inappropriate input wind data [19–21]. Near-surface wind speeds measured on land (but near the shore) are generally lower than those at sea [22,23]. Using offshore wind measurements without correcting for this systematic difference will thus overestimate potential transport rates and deposited volumes. The foredune will additionally alter the regionally representative (on land) wind speed and direction. Offshore directed winds may reverse on the upper beach and the seaward side of the foredune to cause onshore transport and foredune growth [20,24], while onshore directed winds are decelerated on the beach, causing wind speeds in the region where aeolian transport is initiated to be lower than the regionally representative wind speed [19,25]. In addition, onshore oblique winds will deflect in the alongshore direction at the foot of the foredune [19,20,26], resulting in alongshore rather than onshore aeolian transport. These topographic effects are strongest when the foredune is high and steep (e.g., scarped). For a Dutch case study with a 22-m high foredune with a 1:2.5 seaward slope, De Winter et al. [19] predicted the potential deposition volume to drop from 86 to 32 $m^3/m/yr$ when including correction factors for the foredune-induced topographical modification of the regionally representative onshore wind. Despite the marked reduction, the corrected potential deposition still exceeded the measured deposition of about 15 $m^3/m/yr$. This illustrates the common assertion that model failure is primarily due to the neglect of factors that reduce the aeolian supply to below its potential value. Surface moisture, an extensively studied supply-limiting factor, impedes the entrainment of sand [27], which can make part of the beach unavailable as a supply source [28,29]. Surface moisture may also increase the downwind distance needed to achieve the potential (maximum) transport rate (critical fetch [18,30]) to a value larger than the available beach-width. In such a case, the amount of sand blown across the landward margin of the beach onto the foredune is thus less than the potential amount. Model failure is indeed most pronounced on narrow beaches, with a width of about 200 to 300 m marking the upper value for which width rather than wind speed dominates the control on long-term aeolian supply to the foredune [14]. Lag deposits [31,32] and algae or salt crusts [33] are other examples of supply-limiting factors.

Data sets of aeolian activity on the beach spanning a wide range of conditions have illustrated that strong onshore wind events are most supply limited [11,12,34–36]. Such events are often associated with rainfall, high waves and hence substantial run-up on the beach and, in particular, elevated water levels because of wave-breaking induced set-up and a storm surge. The flooding of the beach will reduce beach-width and consequently reduce the transport at the beach-dune transition to well below the potential maximum. When the water levels are high enough, the result will not be deposition on the foredune but rather foredune scarping and erosion. Despite the fact that strong wind events are rather uncommon, the supply limitation may be so strong that they constitute the primary condition under which most of the over-prediction of foredune growth with a potential transport equation arises [11,34,35]. Nonetheless, the reasons how a surge partially or fully shuts down the aeolian system are not entirely understood. In addition to reducing beach-width, a surge also elevates the ground water level in the upper beach for several days [37] with potentially large but unexplored effects on spatio-temporal surface moisture dynamics and hence aeolian processes during and after the surge. Collecting data relevant to aeolian processes during a surge still poses a challenging task. Probably because of a lack of suitable data, recent model improvements to predict aeolian sand transport to the foredune in supply limited situations [18,38,39] have, with the exception of Cohn et al. [40], not been applied in detail to surge conditions. Cohn et al. [40]'s model indicated that run-up under certain conditions with elevated water levels can contribute to foredune growth rather than to scarping and erosion.

The overarching aim of this paper is to further our understanding of aeolian supply limitations imposed by a storm surge. To that end, we first analyze observations of cross-shore surface moisture dynamics under storm-surge conditions collected at Egmond beach, the Netherlands, during the Aeolex-II campaign in October 2017. The storm surge elevated the mean water level to about 1 m above astronomical tidal level, with the largest swash motions during the high-tide storm peak just

reaching the beach-dune transition. We then quantify the surge-induced consequences for the aeolian supply to the foredune with the recently developed fetch-based model of Hage et al. [39] and examine the reasons for the predicted supply limitations (e.g., beach-width; moisture dynamics) with several model scenarios.

## 2. Methodology

### 2.1. Observations

#### 2.1.1. Study Site

The field site was located approximately 3 km south of Egmond aan Zee, the Netherlands in the immediate vicinity of beach pole 41 (Figure 1). The coast at Egmond aan Zee is approximately North-South orientated (7° clockwise rotated with respect to North) and predominantly exposed to North-sea generated wind waves [5,41]. The intertidal beach at Egmond has a mild slope (typically, 1:40) and often contains a single slipface ridge [42], which are typical beach characteristics for this part of the Dutch coast [43,44]. At low tide, the beach is up to 100 m wide. The established foredune is 20–25 m high, has a steep (1:2.5) seaward-facing slope and, from a height of about 10 to 15 m above mean sea level (MSL), is densely covered in European marram grass (*Ammophila arenaria*) [45]. At a slightly larger height, the slope of the foredune lessens substantially toward the crest. This abrupt change in slope marks the location to which storms have previously eroded the foredune by means of scarping or rotational failure [5,46].

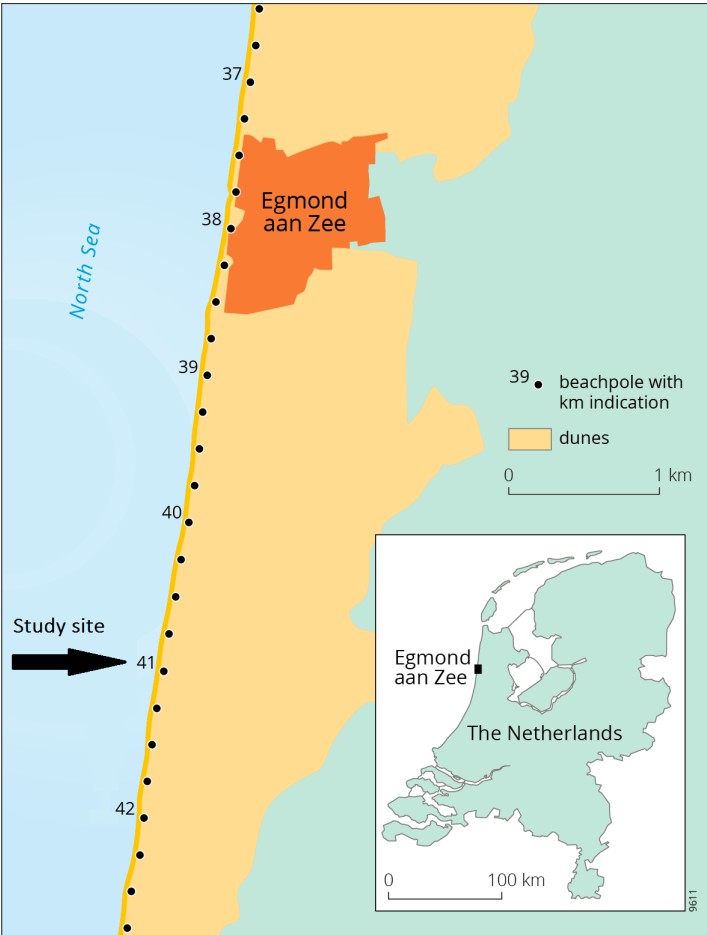

**Figure 1.** Location of study site. The number of a beach pole is the distance (in km) to the zero point at the northern end of the Dutch Holland coast.

The climate in the Netherlands is temperate humid with strong seasonal variability. The storm season is from October to February, with winds predominantly from Southwest to Northwest (i.e., onshore oblique). The monthly mean offshore significant wave height $H_{m0}$ in the storm season is about 1.8 m, which is substantially higher than in the summer months (0.9 m) [41]. During northwesterly storms $H_{m0}$ can increase to over 7 m. The tide is semi-diurnal with an average range of about 1.6 m but water levels vary with spring-neap cycles and wind conditions. During storm surges, which are most prominent during northwesterly storms, the beach may be flooded all the way to the foredune.

### 2.1.2. Aeolex-II

The data were collected in the framework of the Aeolex-II campaign, a field study on aeolian processes executed from 3 to 30 October 2017 by the Department of Physical Geography at Utrecht University. The overarching aim of the campaign was to collect data to investigate how properties of the wind, beach morphology, vegetation and sand (e.g., moisture content) control the timing and amount of aeolian sand transport toward and across the foredune. Measured variables comprised sea water level, groundwater elevation, ambient air temperature, in- and outgoing solar radiation, relative humidity, air pressure, rainfall, wind speed and direction, surface moisture content, beach and dune morphology, saltation intensity, aeolian mass flux and, finally, vegetation type and cover. Here a subset of the measured variables is used, as detailed in Section 2.1.3 below. Other parts of the Aeolex-II data were analyzed in, for example, De Winter et al. [19] and Schwarz et al. [45].

During Aeolex-II the offshore significant wave height $H_{m0}$ peaked at 4 m on 5 October and almost 7 m on 29 October (Figure 2a), associated with gale-force (10-minute mean wind speeds of about 20 to 25 m/s) northwesterly to northern winds (Figure 2b,c). The first wind event took place during spring tide (Figure 2d) and caused minor dune erosion north of the study site, while the second event was during neap tidal conditions and no dune erosion was observed. Both events were associated with a storm surge of about 1 m (Figure 2e). From around 13 to 19 October fair weather conditions prevailed, with low wind speeds ($< \approx 10$ m/s) and small significant wave heights (mostly < 1 m). All wave and water level data are from stations near IJmuiden, to the south of the study site, operated by the Dutch governmental organization Rijkswaterstaat; see Ruessink et al. [46] for precise locations. The wind data (at 10 m above ground) are from the meteorological station IJmuiden (WMO number 06225) operated by the Royal Netherlands Meteorological Institute (KNMI), located some 15 km to the south of the study site on a harbor mole [46].

Atmospheric variables that may influence surface moisture were measured during Aeolex-II with a meteorological station mounted to a beach pole on the dry beach at the field site. An overview of some meteorological variables collected during the field period is given in Figure 3. The air temperature (Figure 3a) was mostly between 10 and 15 °C. The exceptionally high temperatures on 15 and 16 October were caused by hurricane Ophelia, which hit Ireland during this period [47] but at the same time, resulted in very calm and warm conditions in the Netherlands. Figure 3b shows the relative humidity. The tipping bucket in the meteorological station on the beach malfunctioned during part of the campaign, and the rain intensity data in Figure 3c are from the meteorological station Wijk aan Zee (WMO number 06257) operated by the KNMI at N 52° 30′19.0″, E 04° 36′11.0″, about 9.5 km south of the study site. Finally, daily reference crop evaporation (Figure 3d), estimated by the KNMI with the Makkink equation [48] for the Wijk aan Zee station, was generally low: less than about 1.5 to 2 mm/day. The lowest values (<0.5 mm/day) correspond to overcast days. The actual evaporation from the beach was likely to be even less because of the strong limiting effect of unsaturated sand on evaporation [49,50].

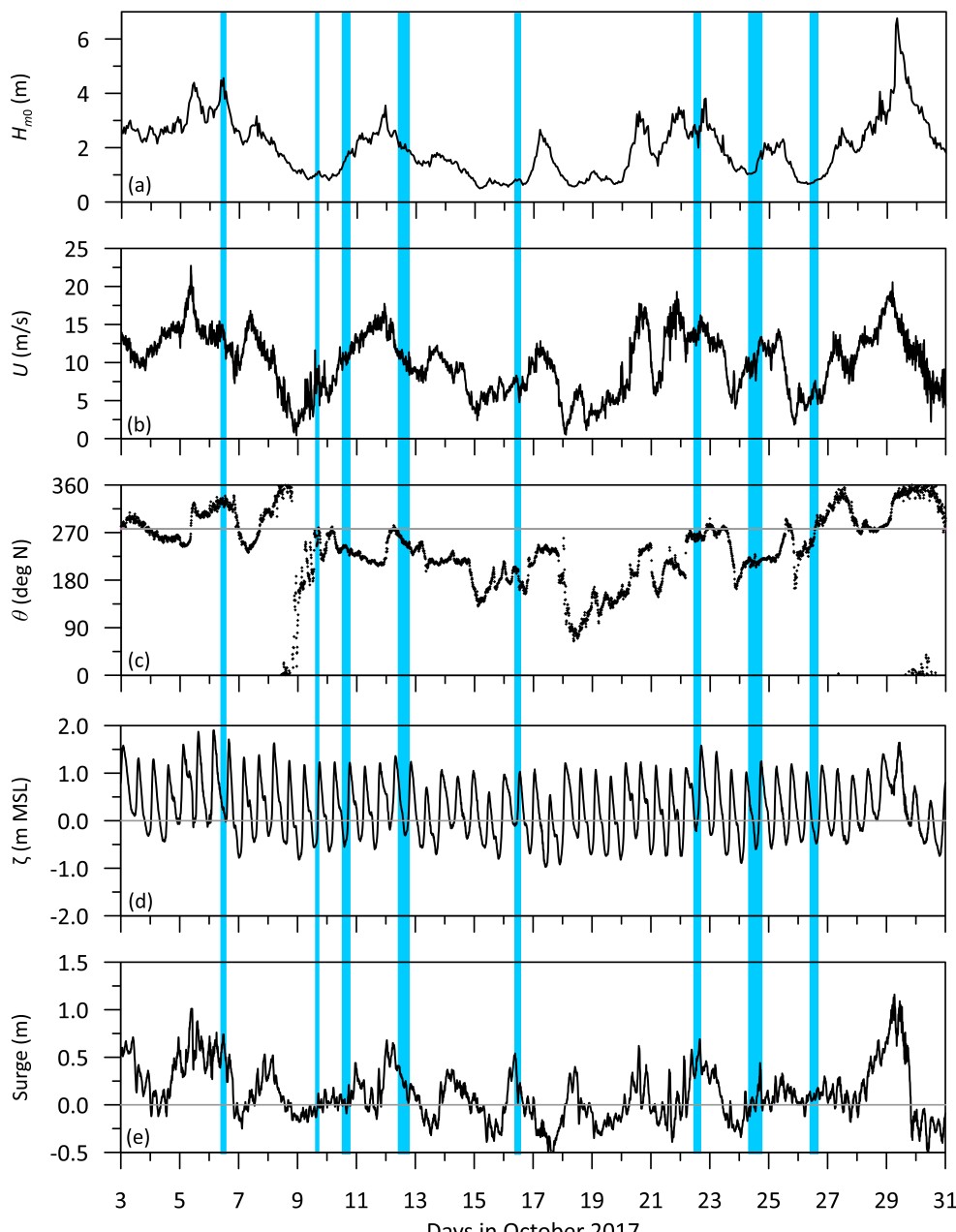

**Figure 2.** Time series of offshore (**a**) significant wave height $H_{m0}$, (**b**) wind speed $U$, (**c**) wind direction $\theta$ with respect to north (N = 0 and 360°, E = 90°, S = 180°, W = 270°), (**d**) water level $\zeta$ with respect to Mean Sea Level and (**e**) surge level during Aeolex-II. The sky-blue vertical bars represent the moments when surface moisture data were collected. The surge level was estimated as the difference between the measured and (predicted) astronomical water level. The horizontal gray line in (**c**) is the shore-normal direction, 277° N).

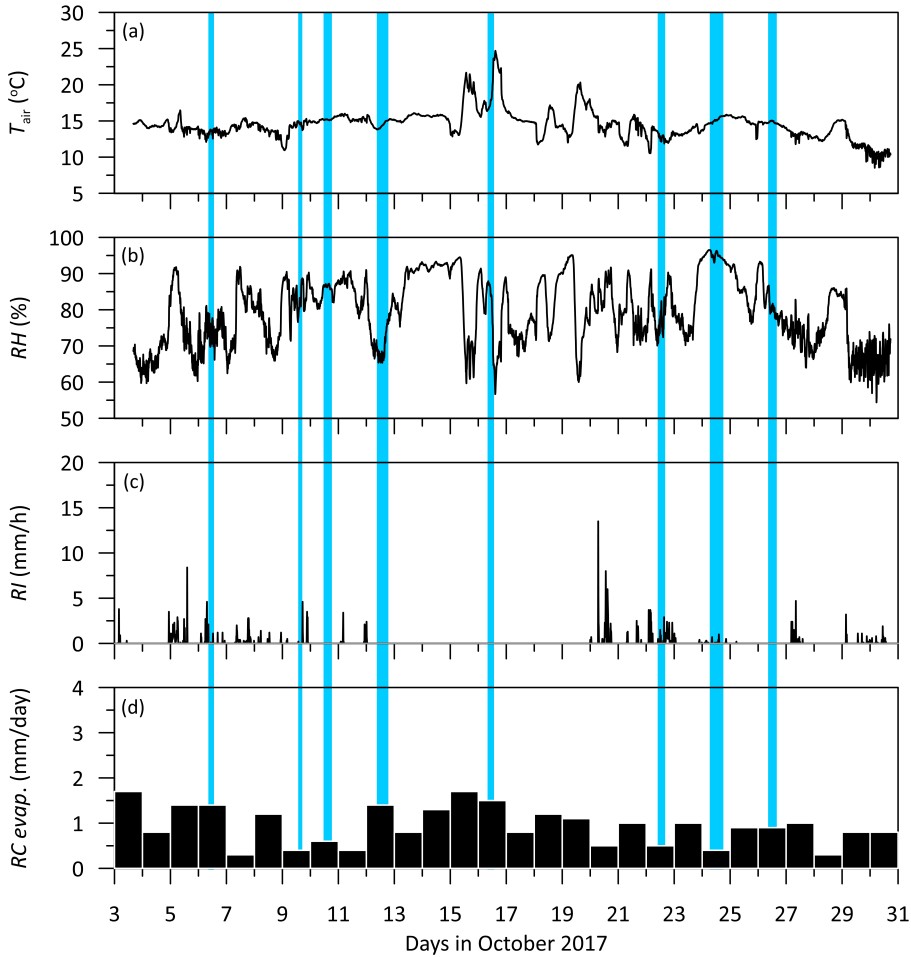

**Figure 3.** Time series of (**a**) air temperature $T_{air}$, (**b**) relative humidity $RH$, (**c**) rain intensity $RI$ and (**d**) reference crop evaporation during Aeolex-II. The data in (**a**) and (**b**) were measured on the beach, those in (**c**) and (**d**) are from the nearby Wijk aan Zee meteorological station operated by the Dutch Royal Netherlands Meteorological Institute. The sky-blue vertical bars represent the moments when surface moisture data were collected.

### 2.1.3. Data Collection and Processing

Surface moisture measurements were conducted using a Delta-T theta probe (type HH2). The probe consists of a read-out unit in a waterproof housing that contains the electronics, which is connected to the sensor via an electric cable. The sensor has 4 stainless steel, 2-cm long rods that are inserted into the sediment. The original rod length is 6 cm but, following [29,51], the pins were shortened to measure the uppermost moisture content that is important to this study. The output of the probe ranges between 0 (dry) and 800 mV (fully saturated). To relate the probe's output to moisture content, a calibration was performed. Beach sediment samples at arbitrary locations on the beach were analyzed with the probe until the entire output range was captured. At each measuring location, a sample was collected with a 2-cm coring ring with the same diameter as the probe. The samples were bagged and immediately sealed for transport. Protocols of standard gravimetric moisture analysis were followed [52]; the wet mass of each sample $m_w$ was determined after which the samples were dried in the oven for 24 h at 105 °C to achieve weight constancy. After drying, the mass of the samples was determined again, $m_d$. The gravimetric surface moisture content $w_s$ [in %] of the sample then is $w_s = 100 \left( m_w - m_d \right) / m_d$. The calibration procedure was followed at the start and end of the of study period and found to give virtually identical results. The relationship between probe output and all $w_s$ was described best with a fourth-order polynomial function, with a correlation-squared $r^2$ of 0.98 and a standard error of 1.17%. The curve clearly revealed that the device is most sensitive to surface

moisture content in the range of 0 to ≈16%. Visual observations in the field showed that full saturation corresponds to $w_s \geq \approx 18\%$. During all measurements in both sampling strategies, the probe was inserted five times into the beach sediment, with the average probe output converted to the surface moisture content using the fourth-order calibration curve.

The moisture measurements were conducted on 6, 9, 10, 12, 16, 22, 24 and 26 October along a cross-shore array of groundwater wells (Figure 4). On 4 October eight wells, termed GW1 to GW8 from sea to land, were placed from the low-tide level to the top of the dry beach with a cross-shore spacing of 10 to 20 m. All sensors were set to store an instantaneous water level value with an interval of 5 min. The positioning of the wells implies that they measured sea water level $\zeta$ when the well was submerged by the tide and ground water level $\eta$ otherwise. GW1–GW4 no longer stood perfectly upright by the end of the first storm. These 4 wells were removed on 7 October, with GW2–GW4 re-installed on 9 October at different cross-shore locations than before the storm. GW2 and GW4 did, however, not collect any data during their second deployment. The recorded time series were processed as detailed in [29,51], resulting in water level series with respect to Dutch ordnance level NAP (about equal to MSL). The wells were removed from the field on 28 October. In addition to moisture measurements at the well locations, measurements were also carried out at markers with an approximately 5-m spacing between the wells. Because these markers had to be removed with every incoming tide and were replaced the next measuring day, their spacing and number varied slightly each measurement day. The marker locations were obtained using an RTK-GPS system after each new transect was placed. Employing the sampling method as previously discussed with five measurements per marker, each location could be handled in about 45 s, implying that the entire cross-shore transect consisting of a maximum of 24 markers during low tide could be handled in less than 20 min. Depending on the inundation level of the beach the entire transect was measured in much shorter time.

A pressure transducer (PT) was deployed approximately 27 m seaward of GW2 (Figure 4). Its data, sampled at 5 Hz, were processed in 5-min. averaged values of sea water level (with respect to MSL), where each block of 5 min was centered around the sample moments of the GWs. Because of its position just landward of the low-tide shoreline, the PT was submerged during the entire tide in case of a storm surge, but during mid and high tide only otherwise. The PT location is used in this paper as the origin of a local cross-shore ($x$) coordinate system, with $x$ positive in the landward direction (Figure 4).

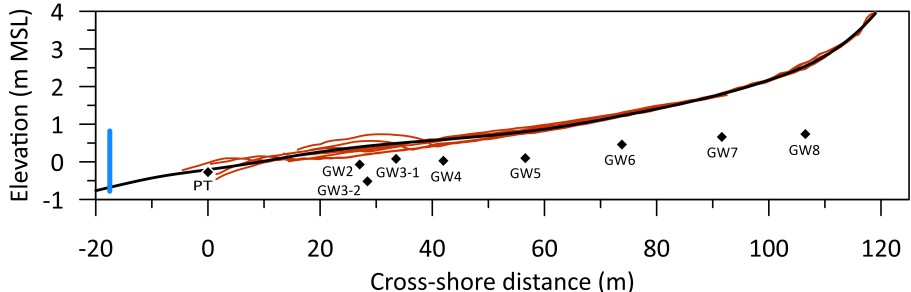

**Figure 4.** Bed elevation $z$ versus cross-shore distance $x$ on all moisture measurement days (six red-brown lines). The thick black line is the campaign-averaged cross-shore profile. Its extension seaward of $x \approx 0$ m is based on a jet-ski survey of the subtidal zone performed on 3 November. Also shown are the locations of the pressure transducer (PT) and the groundwater wells that operated during at least part of the campaign. GW3-1 and GW3-2 are GW3 prior to 7 October and after 9 October, respectively. See text for further explanation. The blue vertical line indicates the mean tidal range.

The cross-shore profile $z(x)$ along the GW-PT array, surveyed with an RTK-GPS, was typically featureless during and immediately after high-wave conditions and contained a bar-trough system in the intertidal zone otherwise. The campaign-average profile, based on 26 individual profiles

(no surveys were performed on 18 and 29 October), was monotonically sloping in the landward direction (Figure 4), with an approximately 1:50 intertidal slope. Additional topographic data were collected before (23 September) and after (3 November) the campaign using an Unmanned Aerial Vehicle (UAV) equipped with a LiDAR system. The two flights covered an area that extended 1.4 km in the alongshore direction, roughly from the mid-tide shoreline to several tens of meter landward of the crest of the foredune. As detailed in Ruessink et al. [46], the collected 3D point clouds were processed into digital elevation models (DEMs) with a $1 \times 1$ m spatial resolution. The volume change of the seaward side of the foredune, taken from the 2.5 m MSL contour to the location of the change in slope [46], was positive (i.e., deposition) in the entire area. The volume gain at the PT-GW array, taken as the alongshore average of a 100-m wide section centered around the array, was about 3.3 m$^3$/m between 23 September and 3 November. This is a quite substantial number for an approximately one-month period, as, for example, the volume gain in the embryo dunes in 2017 as a whole (based on surveys in January and December 2017 [46]) was about 15 m$^3$/m. The median grain size *D* at the array, determined by sieving sand samples collected at the end of the campaign, decreased in the landward direction from about 290 μm near the low-tide level to about 250 μm at the dune foot.

Finally, to aid in the interpretation of the GW data, we determined how each well was positioned with respect to the sea-land transition at any particular moment in time during Aeolex-II by calculating the total water level (TWL) on the beach. As in Cohn et al. [53], the TWL was taken as the sum of the offshore water level (tide and storm surge), the breaking-induced set-up and the 2% exceedance value of the swash. Here, the TWL was computed using the measured IJmuiden water level (Figure 2d), and the empirical Stockdon et al. [54] predictors for set-up and swash with the IJmuiden wave data (e.g., Figure 2a). As can be seen in Figure 5, the TWL peaked at 2.5 m +MSL during two successive high tides on 5 and 6 October, indicating that the swash may just have reached the most landward groundwater well (GW8) during the first storm. Following both peaks, the TWL dropped to the level of GW3 to 5 ($\approx$0.5–0.7 m +MSL). For most of the campaign, however, GW7 and GW8 were above the high-tide TWL and all GWs were above the low-tide TWL. The maximum TWL ($\approx$2.75 m +MSL) during Aeolex-II was reached on 29 October, but by this time all groundwater wells had, as aforementioned, already been removed from the beach.

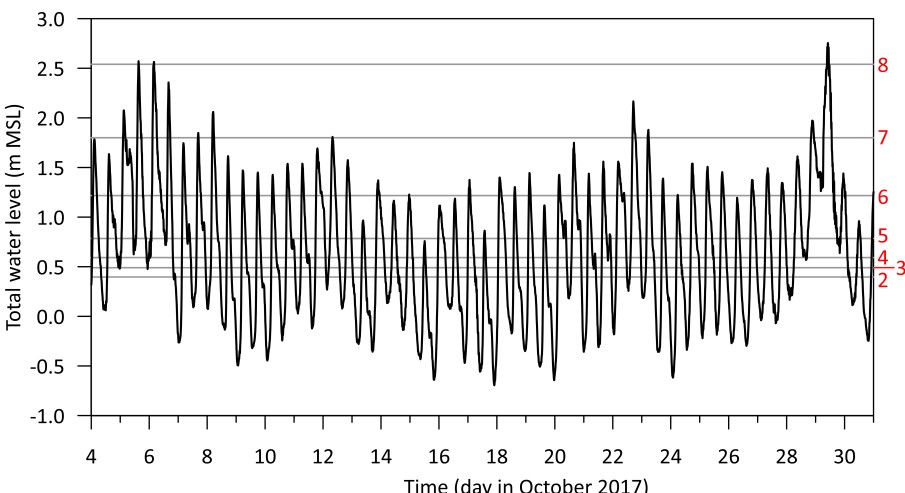

**Figure 5.** Total water level TWL versus time *t*. The horizontal lines are the bed level at the groundwater wells, indicated with the red numbers next to right vertical axis, based on the campaign-averaged cross-shore profile. GW3 is GW3-1. The line for GW3-2 is not shown separately as it is about equal to that for GW2, see Figure 4.

## 2.2. Model

### 2.2.1. Description

The model, called Aeolus, predicts the actual and potential cross-shore aeolian transport rate at the beach-dune transition. Aeolus comprises three modules: (1) a groundwater module to predict cross-shore and temporal water table fluctuations using the one-dimensional (cross-shore) non-linear Boussinesq equation extended with a run-up infiltration, (2) a moisture module that predicts surface moisture content from the predicted depth of the water table and a soil–water retention curve under the assumption of hydrostatic equilibrium, and (3) a fetch-based aeolian transport module in which the spatio-temporally varying surface moisture content is used as a supply-limiting factor. The groundwater and moisture module have been detailed in Brakenhoff et al. [29] and the full Aeolus model has been described in Hage et al. [39]. The Aeolus version used here differs from the version in [39] in that three modifications to the aeolian transport module were made: (1) the threshold for motion is computed explicitly rather than provided as a user-specified input parameter, (2) the scheme to relate surface moisture to the critical fetch has been simplified to equal that proposed in [18] and (3) the cosine-effect [12,13] has been included to facilitate the computation of foredune growth. In the following we provide the main equations of Aeolus.

The cross-shore and temporal water table fluctuations $\eta'$ follow from

$$\frac{\partial \eta'(x,t)}{\partial t} = \frac{K}{n_e} \frac{\partial}{\partial x} \left\{ [d_a + \eta'(x,t)] \frac{\partial \eta'(x,t)}{\partial x} \right\} + \frac{U_l}{n_e}. \tag{1}$$

Here, the prime indicates a predicted value, $t$ is time, $d_a$ is the (constant) aquifer thickness, $K$ is the hydraulic conductivity of the sand, $n_e$ is the (non-dimensional) specific yield and $U_l$ is the run-up infiltration flow rate per unit area [55]. The seaward boundary condition is a moving shoreline at location $x_{sh}(t)$ with elevation $\eta'_{sh}(x_s,t)$

$$\eta'_{sh}(x_{sh},t) = \zeta_0(t) + \xi'_{sh}(t), \tag{2}$$

where $\zeta_0$ is the offshore water level (Figure 2d) and $\xi'_{sh}(t)$ is the breaking-induced set-up at the shoreline, for which the Stockdon et al. [54] parameterization is taken. The imposed landward boundary condition is $\partial \eta'/\partial x = 0$. The ground water module can handle monotonically increasing (in the landward direction) bed profiles only (i.e., no sandbar/trough profiles).

The depth of the water table beneath the bed surface, $h'(x,t) = \eta'(x,t) - z(x)$, is used to predict the surface moisture content $w'_s(x,t)$ using the Van Genuchten [56] soil–water retention curve,

$$w'_s(x,t) = w_{res} + \frac{w_{sat} - w_{res}}{\left[ 1 + (\alpha h'(x,t))^n \right]^{1-1/n}}. \tag{3}$$

Here, $w_{res}$ and $w_{sat}$ are the residual and saturated gravimetric water content, respectively, and $\alpha$ and $n$ are free parameters. The use of Equation (3) implies an immediate response of surface moisture content to water table variations, without hysteresis effects [57]. Earlier work [29,51] has shown that this is a justified approach for the sediments at our field site.

Hage et al. [39] coupled the groundwater and surface moisture modules to a fetch-based aeolian sand transport model, thereby changing the conceptual fetch framework of Bauer and Davidson-Arnott [13] into a predictive model and generalizing the earlier predictive model of Delgado-Fernandez [18] that relied on a video-derived (i.e., observed) and cross-shore constant surface moisture content. Because $w'_s$ decreases in the duneward direction from $w_{sat}$ near the shoreline to $w_{res}$ on the dry beach [29], Hage et al. [39] computed the downwind increase in the aeolian sand transport rate $q$ (in kg m$^{-1}$ s$^{-1}$)

on each output time step of the moisture module with a spatially forward-stepping (i.e., from sea to land) equation,

$$q(i) = \begin{cases} \min\left[q_p, \; q(i_1 - 1) + q_p \sin\left(\frac{\pi}{2}\frac{F(i)}{F_c(i)}\right)\right] & \text{if } w_s(i) \le w_{s,\max} \\ 0 & \text{otherwise,} \end{cases} \tag{4}$$

where $q_p$ is the potential aeolian transport rate. For each grid point with $w_s'$ less than a user-specified moisture threshold $w_{s,\max}$ (typically, 10% [18]) above which aeolian transport is inhibited, the critical fetch $F_c$ is computed as

$$F_c = p(w_s') \times (4.38U - 8.23). \tag{5}$$

The rightmost term in this equation is the wind-speed $U$ dependence of the critical fetch for dry sand as proposed by Delgado-Fernandez [18]. The term $p(w_s)$ describes the moisture dependent increase in the dry sand $F_c$, for which, as in Delgado-Fernandez [18], a step-function is adopted with $p = 1$ for $w_s' < 4\%$, $p = 1.25$ for $4\% \le w_s' < 6\%$ and $p = 1.75$ for $6\% \le w_s' \le 10\%$. Please note that [39] used a slightly different $p(w_s')$, which demanded the rounding of $w_s'$ to multiples of 0.5%; in the present scheme, this rounding is no longer necessary. In Equation (4) we denote a group of consecutive grid points with the same $F_c$ by $i$; in general, there will be 3 $F_c$ groups from sea to land, namely a group with $6\% \le w_s' \le 10\%$ ($i = 1$), with $4\% \le w_s' < 6\%$ ($i = 2$) and with $w_s' < 4\%$ ($i = 3$). In each group, the fetch $F$ is computed as the distance downwind of the last grid point seaward of the start of the group. This is indicated by the $i_1 - 1$ in Equation (4), with $i_1$ the first location index of the $i$th $F_c$ group. The present $F$ computation implies that $F$ is reset to 0 each time $F_c$ changes; that is, at $w_s' = 6$ and 4%. In case $F$ exceeds $F_c$, $F = F_c$ is adopted. In essence, Equation (4) implies that the transport rate $q$ in each group $i$ thus equals the transport rate at its seaward boundary ($= q(i_1 - 1)$ in Equation (4)) increased with the fetch-based trend within the group ($= +q_p \sin((\pi/2)(F(i)/F_c(i)))$ in Equation (4)), up to a maximum of $q_p$.

The potential aeolian transport rate $q_p$ (in kg m$^{-1}$ s$^{-1}$) is predicted with Hsu [58] (see also Davidson-Arnott and Law [12]),

$$q_p = [-0.47 + 4.97D_{mm}]\left(\frac{U_*}{\sqrt{gD}}\right)^3 \times 10^{-5}. \tag{6}$$

The term $[-0.47 + 4.97D_{mm}]$ contains the dependence of $q_p$ on the grain size $D$, which has to be specified in mm (hence, $D_{mm}$). The term between the large brackets is a Froude number, where $g = 9.81$ m s$^{-2}$ is gravitational acceleration. Under the assumption of a logarithmic velocity profile, the shear velocity $U_*$ is related to $U$ as $U_* = aU$, where $a$ depends on the measurement height of $U$ and on the roughness length $z_0$. The Hsu [58] equation does not contain a threshold $U_*$ for motion ($U_{*t}$). Following Delgado-Fernandez [18], Hage et al. [39] applied Equation (6) only when the wind speed exceeded a user-specified threshold value; otherwise, $q_p = 0$. Here, we estimated $U_{*t}$ using Equation (7) and applied Equation (6) only when $U_*$ exceeded $U_{*t}$. We adopted the Shao and Lu [59] formulation, modified following [60] to include the bed slope ($\beta$) effect on the initiation of motion,

$$U_{*t} = \Phi(\beta)\, A_N \sqrt{\frac{\rho_s - \rho_a}{\rho_a}gD + \frac{\gamma}{\rho_a D}}, \tag{7}$$

with

$$\Phi(\beta) = \sqrt{\cos\beta + \frac{\sin\beta}{\tan\Psi}} \tag{8}$$

and $\Psi \sim 33°$ the angle of repose. The non-dimensional parameter $A_N$ is 0.111, $\rho_a = 1.25$ kg m$^{-3}$ is the density of air, and $\gamma$ is $2.9 \times 10^{-4}$ N/m$^{-1}$ [61].

The computation of $q$ in Equation (4) is continued up to and including the beach-dune transition. The $q$ at this transition is henceforth referred to as $q_{bdt}$. Its cross-shore (onshore) component, $q_{bdt,on}$, is (cosine-effect)

$$q_{bdt,on} = q_{bdt} \cos \theta_{SN}, \tag{9}$$

in which $\theta_{SN}$ is the wind direction with respect to the shore-normal direction. The volume growth of the foredune $\Delta V$ during a given time interval can be computed as

$$\Delta V = \frac{\sum (q_{bdt,on}(t)\Delta t)}{(1-p)\rho_s}. \tag{10}$$

Here, $\Delta t$ is the model output time step, $p$ is bed porosity (typically, 0.4) and $\rho_s$ is the sediment density (for quartz sand, $\rho_s = 2650$ kg m$^{-3}$). The computation of $\Delta V$ implies that all sand that crosses the beach-dune transition is deposited on the foredune. This is a realistic assumption for the present study site because of the dense vegetation on the upper part of the foredune [45,46,62,63].

### 2.2.2. Set-Up

The settings of the groundwater module were taken from Hage et al. [39] ($K = 4.63 \times 10^{-4}$ m/s, $n_e = 0.3$ and $d_a = 15$ m), which, in turn, were based on earlier modeling work for the study site (e.g., [64]). The seaward boundary condition was taken as the sum of the offshore water level shown in Figure 2d and the set-up computed with the Stockdon et al. [54] predictor using the offshore wave data collected at IJmuiden (see Figure 2a for the wave height). The campaign-averaged cross-shore profile (Figure 4) was used as the (time-invariant) cross-shore profile $z(x)$, with a cross-shore grid spacing of 0.5 m. The time step in the groundwater module was set to 2 s, with cross-shore profiles of water table fluctuations $\eta'(x)$ written as output every 10 min (i.e., $\Delta t = 600$ s). The settings for the surface moisture module ($w_{res}$, $w_{sat}$, $\alpha$ and $n$) will be based on the fitting the soil–water retention curve, Equation (3), to the Aeolex-II observations as presented in Section 3.1 below. The model was run from 1 September to 4 November 2017, and thus encompasses the entire period between the two UAV-LiDAR surveys. The approximate 3-week period prior to 23 September served as model spin-up time.

The fetch-based aeolian transport module was forced with the wind data measured at IJmuiden (Figure 2b,c), with onshore winds modified to local (i.e., on the beach) conditions to allow for the effect of the high and steep foredune on the wind field. De Winter et al. [19], a study largely based on the Aeolex-II wind data, illustrated that the wind speed on Egmond beach corrected to the same height as at IJmuiden (10 m) was substantially lower than that measured at IJmuiden. The ratio of the local to regional (IJmuiden) wind speed was lowest ($\approx 0.7$) for onshore winds to increase gradually to near 1 for alongshore winds (Figure 6a). De Winter et al. [19] further indicated that the wind direction on the beach was essentially equal to that measured at IJmuiden. In contrast, the wind at the beach-dune transition was often more alongshore directed (Figure 6b), with the deflection angle being largest $\pm(10$–$15°)$ when the regional wind approached the beach at about $\pm 45°$. Therefore, in the computation of the fetch (i.e., in Equation (4)) we used the regional wind direction, but the alongshore deflected direction in the computation of the onshore component of the aeolian transport rate at the beach-dune transition (i.e., in Equation (9)). When the IJmuiden wind was blowing offshore, $q_p$ was set to 0. This implies that we ignore the contribution of offshore winds to foredune accretion due to wind flow reversal on the seaward side of the foredune [20,24]; extensive video observations of Egmond beach did not reveal any noteworthy aeolian activity under offshore winds [65], hence our choice to restrict the Aeolus model to onshore wind conditions. The grain size $D$ was taken as 250 μm, and the moisture threshold $w_{s,max}$ as 10%. Based on an analysis of TWL values for Egmond between January 2015 and 2019 [46], $z_{bdt} = 2.5$ m +MSL was adopted. Finally, extensive observations during Aeolex-II with a vertical array of 5 cup anemometers located just above the high-tide level revealed a typical roughness length $z_0$ of about $1 \times 10^{-4}$ m. Therefore, the ratio $a$ between the wind shear velocity $U_*$ and wind

speed $U$ at $z = 10$ m above ground, was set to $\kappa \log(z/z_0) = 0.0356$, using 0.41 for the Von Karman constant $\kappa$. This $a$ is about 11% lower than the commonly applied $a = 0.04$ proposed by Hsu [66].

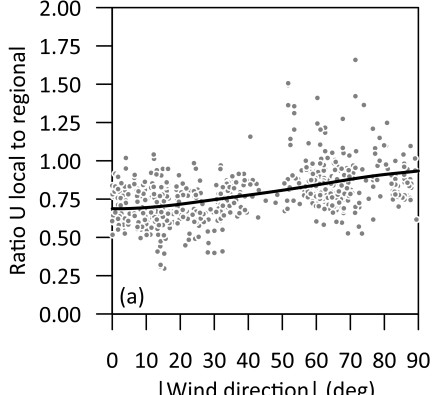 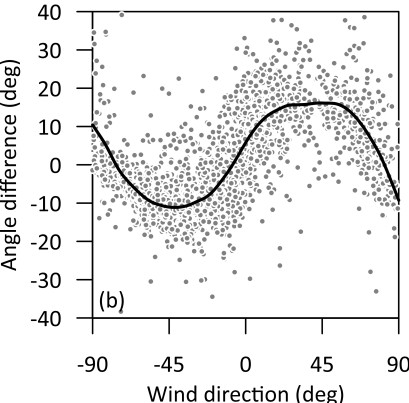

**Figure 6.** Correction of regional (IJmuiden) to local (Egmond beach) wind data. (**a**) shows the ratio of local to regional wind speed versus the absolute value of the regional wind direction with respect to the shore normal, $|\theta_{SN}|$. Local wind speeds were measured at 0.9 m above the bed and were corrected to the same height above ground (10 m) as where the regional values were measured, assuming a logarithmic velocity profile and a roughness length $z_0$ of $1 \times 10^{-4}$ m. The panel contains all 730 observations (dots) taken on the intertidal beach during Aeolex-II and the smooth trend line. (**b**) shows the observed (dots; Aeolex-II) difference between the wind direction at the beach-dune transition and at IJmuiden versus the regional wind direction with respect to the shore normal, $\theta_{SN}$. The line is the smooth trend line. This panel contains about 2950 observations. Shore-normal onshore wind (i.e., coming from 277 °N) has $\theta_{SN} = 0°$; negative and positive angles on the $x$-axis in (**b**) represent winds coming the southwest and northwest, respectively. Because the local to regional wind speed ratio (**a**) was symmetric with respect to $\theta_{SN}$, the absolute value of the direction was used to better constrain the trend line. The angle steering in (**b**) was not perfectly symmetric, perhaps because of the embryo dunes to the south of the instrument array [45]. See [19] for details of data collection and processing. We note that the results in (**a**) are different from those in [19] in the sense that they provided correction values in 45° bins.

## 3. Results

### 3.1. Observations

We first focus our attention on the measurements on 6 October (Figure 7), which were performed after the peak in the storm surge on the previous day and night (Figure 2e). The PT recorded a high-tide water level of 1.9 m +MSL around 03:45 (Figure 7c). The estimated TWL of 2.5 m (Figure 5) indicates that as aforementioned, the largest swashes may just have reached GW8 during this high tide. The groundwater at GW8 peaked near 2.2 m +MSL (Figure 7c), which was about 0.7 m higher than recorded on 4 October before the storm. GW7 was submerged during this high tide (the water level was above the bed elevation at GW7; Figure 7c). After GW7 had emerged near 08:00, the ground water level gradually fell to about 1.5 m +MSL, i.e., to about 0.25 m beneath the bed surface. The water level at GW6 remained constant with time at about 1.2 m +MSL after emergence; this water level equals the bed elevation at GW6. Similar behavior is also seen at the GWs lower on the beach. These constant water level values indicate the presence of a seepage face, which was also visually observed during the measurements as a glassy beach surface. A slight drop in water level at GW6 around 12:00 probably indicates that the point where the water table intersected the beach face moved seaward past GW6 around this time. Because the lower GWs were temporarily removed from the field, we cannot say for how many tides the water table and the (sea) tide remained decoupled after the surge of 6 October. The water levels at GW3-2 remained constant at beach surface elevation (when not submerged by the tide) between 9 and 12 October (not shown), after which they reduced until the submergence by the next tide (e.g., see Figure 8c below). The moisture measurements were performed where the beach was

not completely saturated, from the dune foot to approximately the location of GW7. As can be seen in Figure 7a, $w_s$ was low (<4%) and constant with time landward of GW8 (i.e., landward of the largest swashes during high tide). Between GW7 and GW8 $w_s$ dropped by approximately 10% during the 4-h measurement period, widening the dry beach by approximately 5 to 10 m to $x = 98$ m. In more detail, the part of the beach with the largest reduction in $w_s$ moved seaward during falling tide, from near $x = 105$ m between 08:30 and 10:15, to $x = 95$ m between 10:15 and 12:42. The remainder of the beach, seaward of GW7 ($z < 1.7$ m), remained fully saturated during the entire measurement period.

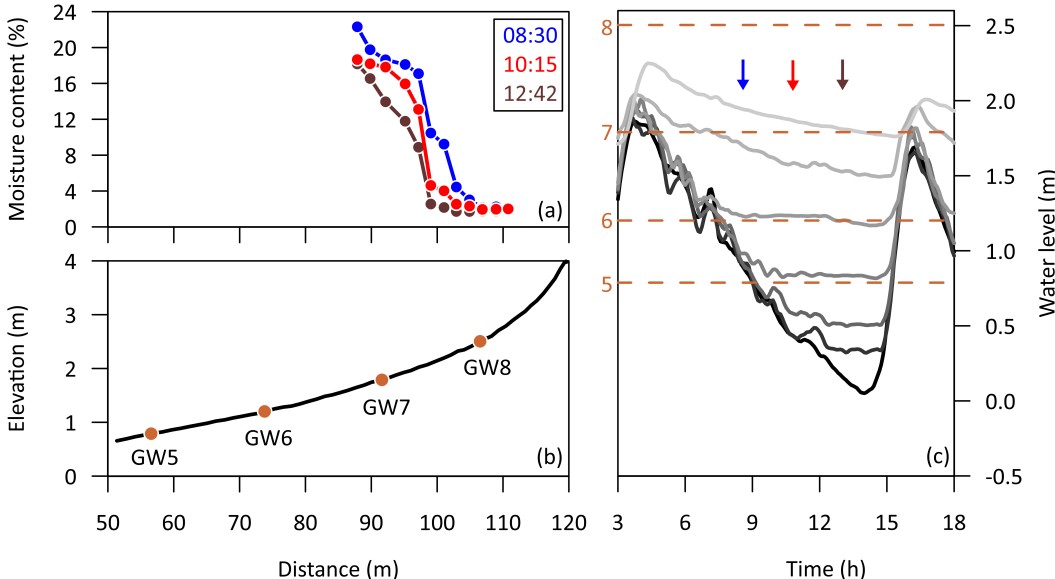

**Figure 7.** (**a**) Surface moisture content $w_s$ at three time intervals during falling tide and (**b**) bed elevation $z$ versus cross-shore distance $x$ on 6 October, at the end of the storm surge that peaked on the previous and night. (**c**) shows time series of water level at six groundwater wells, GW8 (light gray) down to GW3 (dark gray). The black line is the water level at the PT. The three vertical arrows mark the moment of moisture sampling, with the same color coding as in (**a**). The four dashed brown lines are the bed elevation at (top to bottom) GW8 to GW5, see also (**b**).

Figure 8 illustrates the surface moisture and water level measurements during a falling tide without a surge (26 October). The maximum water level recorded by the PT during the preceding high tide was just below 1 m +MSL (Figure 8c) and the TWL estimated for this moment (Figure 5) suggests that the upper limit of the swash zone was near GW6. The water levels at GW8 and GW7 were substantially lower than on 6 October and barely showed any variation with the tide. The surface moisture content at GW7 and GW8 was below 4% and did not vary with time (Figure 8a). At the start of the measurements, just after high tide, $w_s$ rose sharply to about 20% within 15–20 m between GW7 and GW6. With time, the intertidal beach dried, with the lowest $w_s$ around the high-tide level (GW5 to GW6) and on the crest of the intertidal bar near $x = 20$ m and with $w_s$ remaining largest in the trough near $x \approx 30$ m (Figure 8a). The reduction in $w_s$ coincided with the falling of the ground water level (i.e., increase in ground water depth) with time of the wells in the intertidal zone (GW3-2, GW5 and GW6). Despite the overall drying, $w_s$ remained above 10% and at many places above 15%. This suggests that even under these conditions most of the intertidal beach remained too wet to sustain aeolian transport.

The $w_s$ observations of all days without a surge confirm the observations of 26 October and further illustrate that under such conditions the beach can broadly be divided into three cross-shore moisture zones (Figure 9a), consistent with earlier studies conducted under low-wave conditions [29,51,67–70]. These zones were here $x <\approx 10$ m (lower beach), $x \approx 10$–80 m (mid-beach) and $x >\approx 80$ m (upper beach). The lower beach, corresponding to the intertidal beach with $z < 0$ m MSL, was always saturated. The mid-beach, which is the intertidal beach with $z > 0$ m MSL, experienced saturated

conditions immediately after emergence during the falling tide. With time, the surface moisture content dropped to about 10%, or to 5% in the presence of a sandbar (here, between $x \approx 15$ and 40 m). The upper beach had mostly dry (0–5%) conditions, especially at the higher parts of the upper beach toward the dune foot. This cross-shore moisture zonation was disturbed severely by the presence of the surge on 6 October. Then, as described in connection with Figure 7, dry conditions were only found in a narrow strip just above the maximum TWL, while most of the remaining (sub-aerial) beach remained saturated because of exfiltrating groundwater. Moisture values dropped to more intermediate values in a 10–15 m wide zone beneath the maximum TWL only.

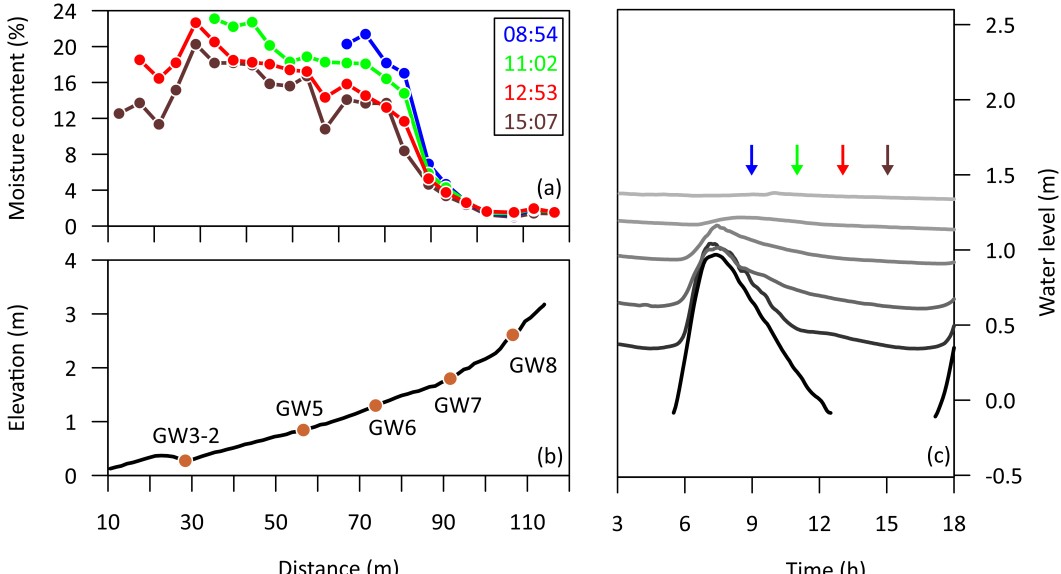

**Figure 8.** (**a**) Surface moisture content $w_s$ at four time intervals during falling tide and (**b**) bed elevation $z$ versus cross-shore distance $x$ on 26 October, a day without a surge. (**c**) shows time series of water level at five groundwater wells, from light to dark gray in seaward direction: GW8, GW7, GW6, GW5 and GW3-2; for their precise location, see (**b**). The black line in (**c**) is the water level at the PT; no data are available at low tide because the PT was not submerged. The four vertical arrows mark the moment of moisture sampling, with the same color coding as in (**a**).

The observed spatio-temporal variability in surface moisture content (Figures 7a, 8a and 9) was, consistent with earlier studies [29,51,57,70,71], controlled largely by the depth $h$ of the groundwater beneath the beach surface. This is illustrated in Figure 10 based on all moisture measurements collected next to a well. When $h$ was less than 0.15 to 0.2 m, the sand remained saturated ($w_s \approx 18$–22%), implying that the capillary fringe still intersected the beach surface. With increasing $h$, $w_s$ rapidly decreased until $h \approx 0.7$ m where the sand reached the field capacity ($w_s \approx 1.5\%$). From this depth, surface moisture content was no longer affected by groundwater depth. The often-used moisture threshold of $w_s \approx 10\%$ above which aeolian transport ceases, corresponds here to $h \approx 0.3$ m. The data from two measurement days are highlighted in Figure 10 to examine potential atmospheric effects. The first day (red dots) is 16 October, when evaporation was expected to be high because of the high air temperature and sunny conditions. However, the $w_s$ values for this day fall within the scatter of the other days, even for large $h$, suggesting that evaporation did not affect moisture content notably. The second day (blue dots) is 22 October, when the measurements were carried out during a light rain. Again, the moisture observations showed the same dependence on groundwater depth as on the other days, except for the observations with $h > 1$ m on the upper beach. Here, $w_s$ was a few percent larger than on the other days. Overall, our results imply that in our data groundwater processes were more important in modulating the surface moisture content than atmospheric processes. The relationship between $h$ and $w_s$ can be described well using Equation (3), see Figure 10. A non-linear fit with

$w_{\mathrm{res}} = 1.5\%$ and $w_{\mathrm{sat}} = 18.2\%$ resulted in $\alpha = 3.18~\mathrm{m}^{-1}$ and $n = 5.47$. This best-fit curve, used in the Aeolus modeling, has a mean square error of about 2.9%.

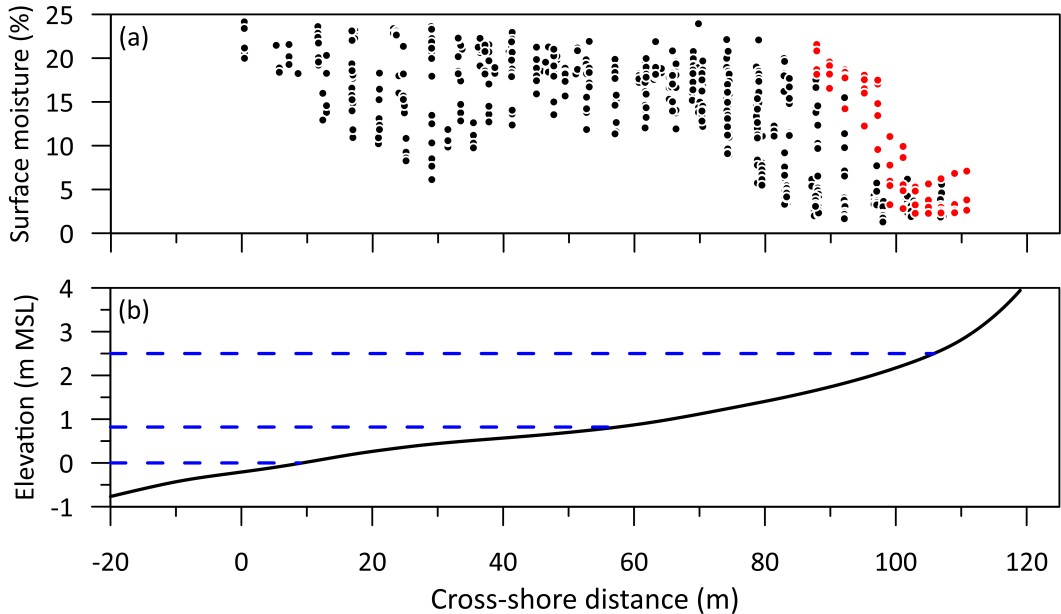

**Figure 9.** (**a**) Surface moisture content $w_s$ versus cross-shore distance $x$ based on all measurement days (848 observations). The observations made on 6 October (immediately after the first storm) are shown in red. (**b**) shows the campaign-averaged cross-shore profile. The blue dashed lines are (from bottom to top) mean sea level, the normal mean high-tide level and the maximum total water level on 6 October.

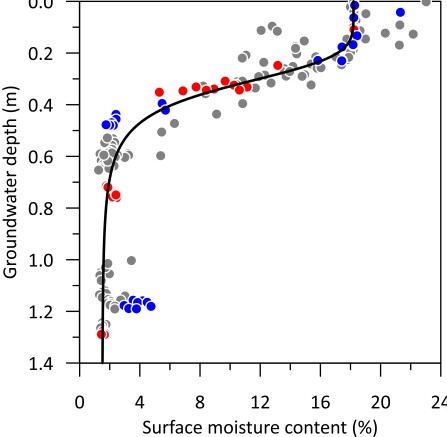

**Figure 10.** Groundwater depth $h$ versus surface moisture content $w_s$. The red dots are the measurements of 16 October, when the air temperature was rather high because of hurricane Ophelia (Figure 3a). The blue dots were obtained on 22 October in a light rain (Figure 3c). The line is the best-fit water retention curve, Equation (3). The total number of observations is 194.

### 3.2. Modeling

Consistent with the observations in Figure 7, the groundwater-moisture model predicted most of the beach to remain saturated after the storm-surge high tide on 6 October (Figure 11; $x < \approx 80$ m, $w_s' = w_{\mathrm{sat}}$). Surface drying was predicted to remain restricted to an approximately 20 m wide zone ($x \approx 80$–100 m). This is slightly wider than in the observations, possibly because of the neglect of groundwater recharge from the dune in the landward boundary condition of the groundwater module [64]. Consequently, the predicted groundwater in the upper beach sat slightly deeper beneath the bed surface and fell faster with time than in the observations. This resulted in a zone with reducing

surface moisture content that extended further seaward from the fully dry zone with time than in the observations. The predicted deeper groundwater also resulted in a further seaward location of the groundwater exit point on the beach, between GW6 and GW5 as opposed to between GW7 and GW6 in the observations. However, in the predictions the groundwater depth at GW6 remained less than 0.2 m and, accordingly, $w'_s$ still equaled $w_{sat}$ (Figure 11). Despite these groundwater-induced discrepancies, the overall spatio-temporal change in surface moisture after the surge is predicted fairly well.

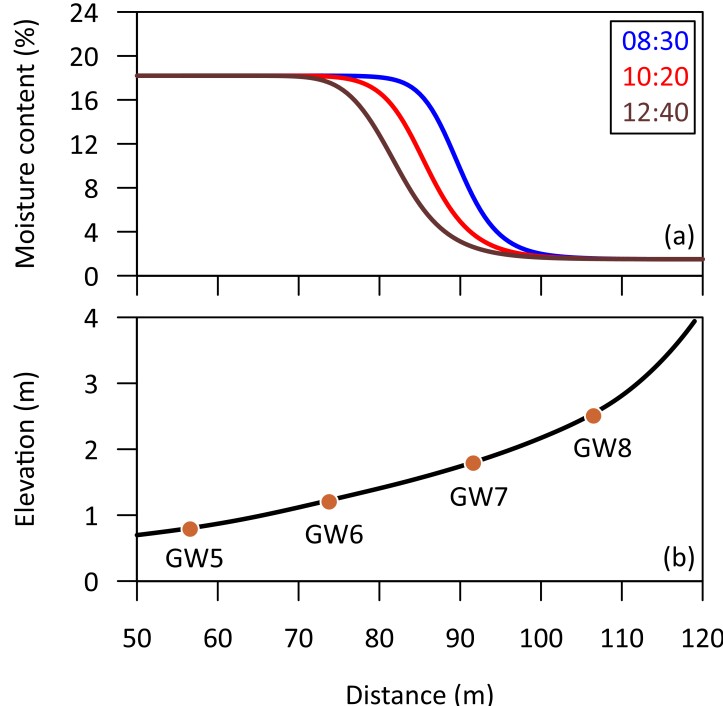

**Figure 11.** (**a**) Predicted surface moisture content $w'_s$ at the three observational time intervals during falling tide on 6 October versus cross-shore distance $x$, compare to Figure 7a. (**b**) contains the campaign-averaged bed elevation $z$ versus cross-shore distance $x$ with the location of the upper 4 GWs. Predictions in (**a**) resulted from predicted groundwater elevations (Equation (1)) and the soil–water retention curve (Equation (3)) with $w_{res} = 1.5\%$, $w_{sat} = 18.2\%$, $\alpha = 3.18$ m$^{-1}$ and $n = 5.47$ (Figure 10).

During the storm on 5 and 6 October the onshore potential transport $q_{p,on}$ followed the wind speed (compare Figure 12a and Figure 12e) and peaked near 0.03 kg/m/s. In contrast, the actual transport $q_{bdt,on}$ varied with the tide and was often considerably lower than $q_{p,on}$ (Figure 12a). During the two highest tides, aeolian transport was predicted to nearly cease ($q_{bdt,on}/q_{p,on} < 0.2$; Figure 12b). In more detail, the ratio $q_{bdt,on}/q_{p,on}$ was lowest just after each high tide. The temporal delay of the high-tide water table in the upper beach with respect to the sea-surface tide caused a delay in wetting of the sand, and hence $w_{s,max}$ was at its most landward position shortly after high tide. The predicted foredune growth $\Delta V$ based on $q_{bdt,on}$ (with $p = 0.4$ and $\rho_s = 2650$ kg/m$^3$) for 5 and 6 October was almost 0.49 m$^3$/m, about 66% of the potential foredune growth of 0.74 m$^3$/m. This substantial reduction highlights the relevance of the storm surge in limiting onshore aeolian sand transport and, hence, foredune growth. On the day before (4 October) and after (7 October) the storm surge the onshore transport at the beach-dune transition was predicted to be virtually equal to the potential onshore transport. Under these conditions the supply was thus primarily controlled by wind characteristics rather than by supply-limiting factors. Only around high tide the transport was slightly supply limited, with $q_{bdt,on}/q_{p,on}$ reducing to 0.7–0.8 (Figure 12b). This is consistent with video observations of aeolian activity at Egmond beach [36], from which a shift from unlimited to supply limited conditions from high to low tide was sometimes inferred. For the period between the two UAV-LiDAR surveys, Aeolus predicted an actual and potential $\Delta V$ of about 4.1 and 4.6 m$^3$/m, respectively. The predicted

actual growth is rather close to (albeit somewhat above) the observed $\Delta V = 3.3$ m$^3$/m in the 100-m alongshore section centered around the instrument array. Possibly, the over-prediction is due to the wider cross-shore zone with drying sand after the surge than in the observations, or to the neglect of other supply-limiting factors such as rainfall.

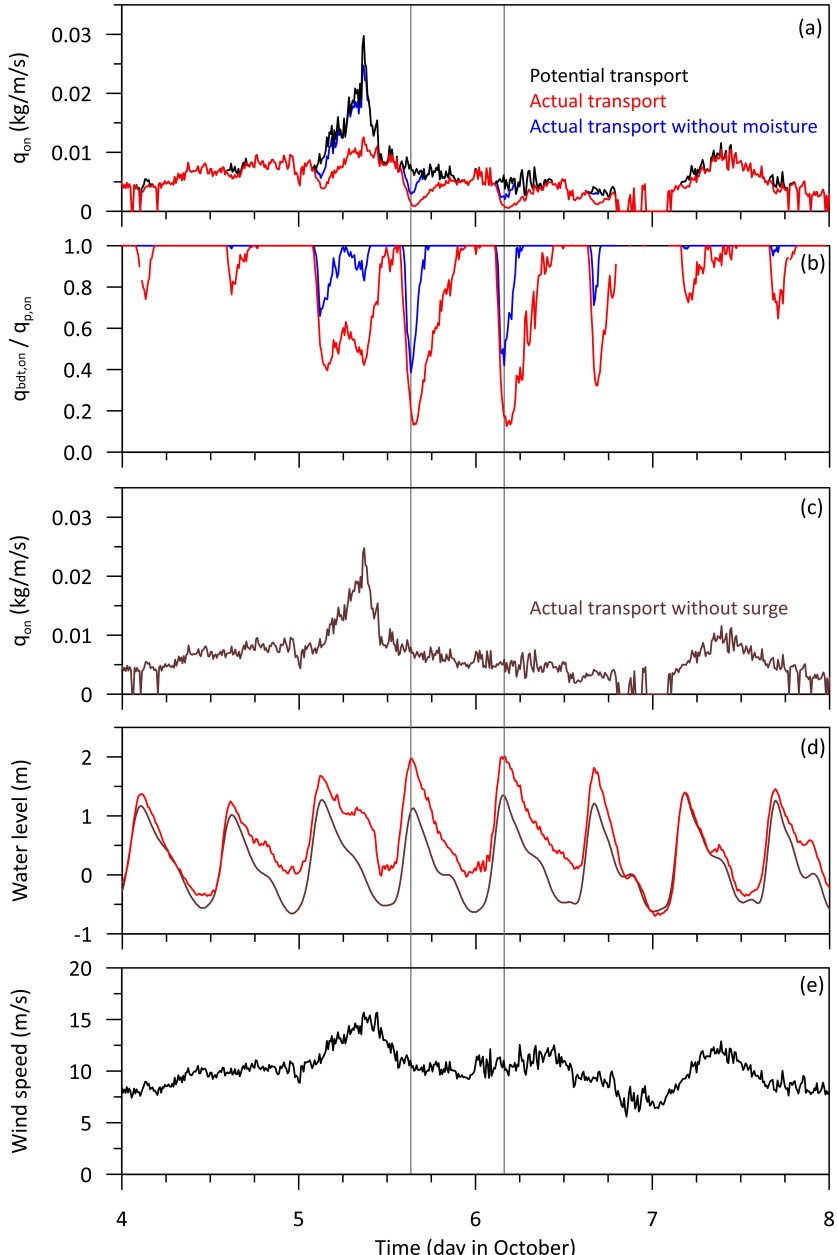

**Figure 12.** Results of aeolian transport modeling for a selected 4-day period in early October 2017. (**a**) shows the onshore potential transport rate $q_{p,on}$ (black line) and the onshore actual transport rate at the beach-dune transition $q_{bdt,on}$ (red line). The blue line in (**a**) is $q_{bdt,on}$ in case the effect of surface moisture on $q$ is ignored (i.e., $w_s = 0$ at all non-submerged locations). (**b**) illustrates the ratios between $q_{bdt,on}$ and $q_{p,on}$ with (red line) and without (blue line) surface moisture. (**c**) shows the results of another scenario (dark brown line) in which the astronomical tide is used as seaward boundary condition in the groundwater module (i.e., no storm surge and run-up; surface moisture is included). The applied astronomical boundary condition in (**c**) is the dark brown line in (**d**), with the red line in (**d**) being the seaward boundary condition applied in both model simulations shown in panel (**a**). The wind speed on the beach is given in (**e**). The two vertical lines through all panels are the two moments with the highest tides on 5 and 6 October.

## 4. Discussion

The predicted $q_{\text{bdt,on}}/q_{p,\text{on}} < 1$ during the surge may have arisen in two complimentary ways. First, the surge reduced beach-width and, consequently, the maximum available fetch. Secondly, the surge caused most of the beach that emerged during falling tide to remain saturated and to not become available for aeolian transport. To disentangle these two effects, we ran two additional simulations, the results of which are also shown in Figure 12. In the first simulation we took the groundwater predictions of the default run (thus including the surge) but we set the moisture content of the emerged beach to 0%. In this way, aeolian transport can commence immediately at the shoreline and the critical fetch is determined by wind speed alone. This run thus shows the effect of the surge-induced surface moisture effects given the surge-induced reduction in beach-width. As can be seen in Figure 12a (blue line), the onshore aeolian transport at the beach-dune transition is now predicted to be much closer to the onshore potential transport. The ratio $q_{\text{bdt,on}}/q_{p,\text{on}}$ reaches less low values at two high tides on 5 and 6 October ($\approx 0.4$ compared to 0.1), is less than 1 for a reduced time interval around high water only, and is no longer delayed with respect to the high tide (Figure 12b). The $\Delta V$ for 5 and 6 October in this simulation was about 0.68 m$^3$/m, 0.19 m$^3$/m larger than in the default run and now 92.5% of the potential $\Delta V$. Further note that transport limitations at high tide were predicted to largely vanish on the other days shown in Figure 12. In the second additional simulation, the groundwater module was run with the astronomical tide as seaward boundary condition (Figure 12d), i.e., without the surge and without waves, but surface moisture was computed for the entire emerged beach. This simulation thus illustrates the effect of the surge-induced reduction in beach-width. We found that without the surge the maximum fetch was nearly always sufficient for $q_{\text{bdt,on}}$ to reach $q_{p,\text{on}}$ at the beach-dune transition (compare Figure 12c to Figure 12a), with a resulting $\Delta V$ for 5 and 6 October that was only 2.4% lower than the potential $\Delta V$. On the whole, these simulations illustrate that the influence of the storm surge in limiting aeolian transport at the beach-dune transition is due to fetch limitations imposed by the reduced beach-width and the saturation of the emerged (intertidal) beach. The low slope (1:50) and low conductivity ($K = 4.63 \times 10^{-4}$ m/s) associated with the fine-medium grain size ($D = 250$ μm) at our site may have favored the large seepage face extent [72] and hence its importance for supply limitation. Future work is necessary to establish the role of seepage face formation and associated saturated moisture conditions in limiting aeolian supply on steeper and coarser-grained beaches.

The model predicted that aeolian supply was unlimited ($q_{\text{bdt,on}}/q_{p,\text{on}} = 1$) under non-surge conditions, except for a restricted amount of time around several high tides. Indeed, the vast majority of the difference between the predicted actual (4.1 m$^3$/m) and potential (4.6 m$^3$/m) deposition over the Aeolex-II study period was due to the storm surges on 5–6 and 29 October. Our model results thus corroborate earlier field studies [11,34,35] that inferred storm surges to be the primary condition under which most of the over-prediction of foredune growth with a potential transport equation arises. Future work is necessary to see whether the inferences we made based on the approximately 1.5-month long study period also apply to seasonal and annual time scales. Model predictions on these larger time scales may reveal how supply limitation depends on surge characteristics, such as surge duration, magnitude and timing with respect to high tide. It is further possible that certain conditions that are not in our data set, such as strong onshore winds without substantially elevated water levels, may also result in fetch and hence supply limitations at our site [36].

Although the reasonable agreement between measured (3.3 m$^3$/m) and predicted (4.1 m$^3$/m) $\Delta V$ is encouraging, also for longer term predictions, there is certainly scope for model improvement. First, the landward boundary condition for the groundwater module could be improved to handle recharge from the foredune. Also, the groundwater module can handle monotonically increasing bed profiles only and, accordingly, the effect of an intertidal sandbar on surface moisture content (Figure 9; [73]) and on aeolian sand transport [74] cannot be predicted. Secondly, the surface moisture module is driven entirely by groundwater fluctuations; meteorological effects, such as evaporation and rainfall, are ignored. For the present data, we found no evidence that evaporation was relevant (Figure 10), but the range of conditions encountered was admittedly small. It is also possible that we

missed evaporation effects because the applied Delta-T theta probe provided moisture content averaged over the top 2 cm of beach sand. The possible drying of the uppermost layer of grains on especially sunny and windy days was thus not measured. The role of rainfall on aeolian transport is unclear. For example, in the transport model of Duarte-Campos et al. [75] aeolian transport is set to 0 when rain intensity exceeds 0.05 m/h, while field measurements reveal non-zero aeolian transport under larger rain intensities [20,76,77]. Also, during Aeolex-II we visually observed substantial aeolian transport during such moderate rainfall, especially when winds were strong. In addition, van Dijk et al. [78] illustrated that severe rain (>2 mm/h) may initiate aeolian transport by splash, although this effect is probably limited to the very early stages of a shower [76]. Thirdly, the Aeolus model assumes a homogeneous sediment. Sand supply limitations often observed under heterogeneous sediments [31,32], typical of nourished beaches and likely caused by the development of a lag deposit, are not included. Fourthly, wind speeds may reduce across the beach [19,25], especially when the wind is cross-shore directed. This variation, which is not accounted for in Aeolus, could influence aeolian transport rates at the beach-dune transition. Finally, it would be useful to include other equations to compute $q_p$ [79,80] to examine the sensitivity of $\Delta V$ predictions to the potential transport equation.

## 5. Conclusions

Our observations at the low-gradient (1:50) beach of Egmond aan Zee, Netherlands, illustrate that spatio-temporal surface moisture content immediately after a surge is controlled by the maximum total water level during the surge peak and the strongly elevated groundwater levels in the upper beach. The latter results in the decoupling of the groundwater from the sea water level toward low tide and hence the development of a seepage face. The surface moisture content is time-invariant and low (<4%; dry sand) above the maximum total water level, while the sand seaward of the groundwater exit point remains saturated with time ($\approx$18%). Drying of beach sand is restricted to a narrow (here, 10–15 m) cross-shore zone between the maximum water level and the groundwater exit point. The beach-width reduction by the surge-induced inundation and the persistent saturation of the emerging beach are both predicted to contribute to fetch limitations that substantially reduce aeolian supply to the foredune, for the 2-day surge period examined here to about 66% of the potential supply. Because during non-surge conditions the supply is almost always equal to the potential supply, our model results provide quantitative support for the common assertion that storm surges are the primary condition that cause the overestimation of measured foredune growth with a potential transport equation. Future work is needed to test the generality of our findings for other beach slopes and longer prediction intervals.

**Author Contributions:** Conceptualization, J.T.T., J.J.A.D. and G.R.; methodology, J.T.T. and G.R.; field investigation, J.T.T.; writing—original draft preparation, J.T.T. and G.R.; writing—review and editing, all authors; visualization, J.T.T. and G.R.; supervision, J.J.A.D., C.S.S. and G.R.; project administration, G.R.; funding acquisition, G.R. All authors have read and agreed to the published version of the manuscript.

**Funding:** This work is part of the Vici research programme "*Aeolus* meets *Poseidon*: wind-blown sand transport on wave-dominated beaches" with project number 13709, which is financed by the Dutch Research Council (NWO).

**Acknowledgments:** We thank Bas van Dam, Marcel van Maarseveen, Henk Markies, Mark Eikelboom and Arjan van Eijk for excellent technical support during Aeolex-II; and Corinne Böhm, Job van Beem and Jorn Bosma for their indispensable help during the measurements and simply a great time in the field.

**Conflicts of Interest:** The authors declare no conflict of interest. The funders had no role in the design of the study; in the collection, analyses, or interpretation of data; in the writing of the manuscript, or in the decision to publish the results.

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
