# Peer review of "Consequences of a Storm Surge for Aeolian Sand Transport on a Low-Gradient Beach"

_jmse, doi:10.3390/jmse8080584_

Round 1

Reviewer 1 Report

This is an interesting paper discussing the impacts of storm surges on predictions of long-term potential foredune growth. Overall the experiment design is sound and the analysis of results is solid. I don't have major concerns but I have some minor suggestions:

  1. The timing of peak surge and astronomical tide seems to be a potential factor which could affect this study's conclusion. Can you add more discussions around how surge-tide interaction would change your conclusion?
  2. This study seems to be very site-specific. Is there any way to generalize your findings?

Author Response

We thank the reviewer for the constructive comments. The comments are repeated below, followed by our response, which is preceeded by **. The line numbers in our response refer to the line numbers in the revised manuscript with all changes in bold face.

The timing of peak surge and astronomical tide seems to be a potential factor which could affect this study's conclusion. Can you add more discussions around how surge-tide interaction would change your conclusion?

** The timing of peak surge with respect to the tide is indeed of influence to the degree of supply limitation. A surge peak coinciding with high tide is likely to limit aeolian transport more than a surge peak with low tide. Often, a surge is of longer duration than the semi-diurnal cycle (e.g., Figure 2), so even during the latter situation the water level at high tide will be affected. Irrespective of the timing, the surge is likely to be the main condition under which aeolian supply to the dunes is limited, because our model results indicate that aeolian transport is unlimited under non-surge conditions (Fig. 12, lines 495-506). We have added to the text in the Discussion (lines 502-504) that modelling on multi-annual time scales is necessary to examine the role of surge characteristics, such as timing with respect to the tide, but also surge height and duration, in determining the degree of supply limitation.

This study seems to be very site-specific. Is there any way to generalize your findings?

** We agree that the precise amount of supply limitation (66%) is specific to the examined surge at our site. However, in our opinion, our conclusion that surges are the dominant condition that limit supply to the foredune is general (lines 495-506). We base this on the correspondence of our modelling results with earlier observations/suggestions of the impact of surges on aeolian supply at other sites (line 499). Additional field observations are necessary to examine the dependence of the degree of supply limitation on, for example, beach slope and grain size (lines 490-494). We expect that the resulting inter-site differences in moisture dynamics are captured by our model because of the physics-based groundwater module. Thus, the model will also be a tool in generalizing the findings to other sites.

Reviewer 2 Report

n/a

Author Response

No comments were provided.

Reviewer 3 Report

Interesting observed data are presented. Some discussions and notations are not well explained/defined. 

  • Both Hs and Hm0 are used for significant wave height.
  • Figure 2: better to explain about the wind direction.
  • explain accuracy of computed TWL shown in Fig.5. Better to separately show the estimated surge height on top of the measured tide.
  • equation (4). clarify the definition of "i". 
  • equation (4): don't you have applicable range of F(i) relative to Fc? Can you capply this when F(i)>Fc?
  • what is i1?
  • better to illustrate the definition of the wind direction.
  • better to show the location, Ijmuiden, where wind was measured.
  • In the caption of Fig.6. Not clear what you meant in "Local wind speeds were corrected to the same height above ground (10 m) as where
    the regional values were measured, assuming a logarithmic velocity profile and a roughness length
    z0 of 1  10?4 m." Did you measure the wind velocity at different elevation and used that data for estimation of U at 10m above the bed based on the assumption of log-profile? How about the difference of the ground level?
  • Clarify the definition of the positive direction of the difference of wind directions between each site and at Ijmuiden. (it is better to illustrate these difference. it is hard to understand)
  • L393-395: it is not convincing from Fig9 to say that the moisture zones can be divided into three.
  • Figure 12. All the qs are based on the emperical model. Please discuss the validity of the predicted qs.

Author Response

We thank the reviewer for the constructive comments. The comments are repeated below, followed by our response, which is preceded by **. The line numbers in our response refer to the line numbers in the revised manuscript with all changes in bold face.

Both Hs and Hm0 are used for significant wave height.

** We thank the reviewer for spotting this inconsistency. The "Hs" was present in the y-axis title and the associated caption of Figure 2a. Both are now changed into Hm0. Hs is no longer used.

Figure 2: better to explain about the wind direction.

** We added wind direction to Figure 2 (new panel c) and briefly explained it in the caption.

explain accuracy of computed TWL shown in Fig.5. Better to separately show the estimated surge height on top of the measured tide.

** We do not have measured TWL for Aeolex-II. However, TWL was based on measured water levels (Figure 2d) and predictors for set-up and swash based on field data (Stockdon et al., 2006). These predictors are well accepted and hence often applied in the international literature. We use the TWL only to indicate how far up the beach marine processes may have acted. The spatio-temporal dynamcis of the observed groundwater levels and the surface moisture are consistent with the TWL estimates, which adds confidence to their accuracy. The surge height and measured tide are already shown together in Figure 2.

equation (4). clarify the definition of "i".

** We modified the text beneath Equation (4) to better explain that "i" is a group of pixels with the same critical fetch (line 283). In the model set-up used in the paper, there are generally 3 groups. These are (from sea to land) the cross-shore locations where the moisture content is between 6 and 10%, between 4 and 6%, and less than 4%. The boundaries of 4 and 6% are due to the way moisture content is related to the critical fetch, Equation (5).

equation (4): don't you have applicable range of F(i) relative to Fc? Can you capply this when F(i)>Fc?

** Yes, we do, and we realize that we had unfortunately omitted this from the text. At all grid points where F exceeds Fc, F is set equal to Fc to avoid unrealistic behaviour of the sin-term in Equation (4). This has been added to the text in lines 286-287. We have also extended the final line of the paragraph beneath Equation (4) to re-state that the transport rate cannot become larger than the potential rate (line 290).

what is i1?

** i1 is the first index of the pixels with the same critical fetch, see line 285.

better to illustrate the definition of the wind direction.

** We suppose this comment refers to Figure 6 and the associated caption. We have extended the caption of Figure 6 to better indicate that the angles on the x-axis of both panels are with respect to the shore-normal direction. As now explained in the caption, a zero-angle implies the wind to blow onshore from 277 deg N (= shore-normal direction); negative and positive angles on the x-axes are from the southwest and northwest, respectively. Also note that the shore-normal direction is now given in Figure 2c in relation to an earlier remark on wind direction.

better to show the location, Ijmuiden, where wind was measured.

** The location of IJmuiden is stated in Section 2 (lines 135-140). We respectfully disagree that an additional figure is necessary. Instead, we now refer to an earlier published data descriptor paper (Ruessink et al., 2019, in the MDPI journal Data) in line 140.

In the caption of Fig.6. Not clear what you meant in "Local wind speeds were corrected to the same height above ground (10 m) as where
the regional values were measured, assuming a logarithmic velocity profile and a roughness length
z0 of 1  10??4 m." Did you measure the wind velocity at different elevation and used that data for estimation of U at 10m above the bed based on the assumption of log-profile? How about the difference of the ground level?

** We have modified the caption of Figure 6 to better explain this. All local measurements were conducted at 0.9 m above the bed and corrected to 10~m (where the regional values were measured) using a log-profile with the stated roughness length z0. The reviewer has correctly interpreted our original text and we hope that the present text is even clearer.

Clarify the definition of the positive direction of the difference of wind directions between each site and at Ijmuiden. (it is better to illustrate these difference. it is hard to understand)

** We now better explain in the caption of Figure 6 what is shown on the x-axis (see earlier point about the wind direction). The vertical y-axis in Figure 6b is the difference between the local and regional wind direction versus the regional wind direction with respect to the shore-normal. For example, if the regional direction is -45 deg, then the difference is about -10 deg. Thus, the local direction is -55 deg, i.e., more alongshore. If the regional direction = +45 deg, the difference is about +15 deg. This implies that the local direction is now about +60 degrees, again more alongshore. We also slightly modified lines 340-341 to clarify this.

L393-395: it is not convincing from Fig9 to say that the moisture zones can be divided into three.

** We agree that the three moisture zones in Figure 9 are difficult to see, compared to similar figures for other sites existing in the literature. However, we hope that with the cross-shore ranges and associated text the zones are reasonably clear, see lines 395-408.

Figure 12. All the qs are based on the emperical model. Please discuss the validity of the predicted qs.

** This is a valid point, as we do not have measurements of qs at the beach-dune transition. Therefore, we converted the total onshore transport during the campaign into a deposition volume (Equation 10) and compared the predicted deposition volume to the measured volume (lines 460-463). Although the predicted volume is larger than the measured volume, the values are comparable. We have slightly modified line 462 to better illustrate the close agreement. We also state various reasons why the predicted volume may be larger than the measured volume (lines 463-465).

Reviewer 4 Report

This work focuses on analyzing and quantitating of storm surge contribution to long-term potential foredune growth predictions on a low-gradient Beach. The object of this study is within the scope of the Special Issue "Storm Tide and Wave Simulations and Assessment" of the JMSE journal. My suggestion is a minor revision.

Minor Comments:

  1. What is the reason for choosing a “low-gradient Beach” as a study case? More details should be provided in the manuscript.
  2. In the present study, only a storm surge event induced by hurricane Ophelia was used. Is an event representative? More details should be provided in the manuscript.

Author Response

We thank the reviewer for the constructive comments. The comments are repeated below, followed by our response, which is preceded by **. The line numbers in our response refer to the line numbers in the revised manuscript with all changes in bold face.

** Our field site, Egmond aan Zee, has been a key location in Dutch coastal research since the seminal work of Battjes and Stive in the 1980s (Battjes-Stive, 1985, Journal of Geophyhsical Research-Oceans). It has hosted several national and international field campaigns. It is representative of many of the North Sea beaches exposed to the high waves from extratroprical storms. Such beaches have typical slopes of about 1:30 to 1:100, and can thus be classified as low-gradient beaches. We added a remark on the representativeness of the site in lines 104 and 105. In addition, to avoid the impression that the aim of our paper, as expressed in the final paragraph of the Introduction section, can only be carried out on a low-gradient beach, we removed "the low-gradient (1:50)" phrase from line 90. Obviously, "low-gradient" is still contained in the paper title, the abstract and the site description. Finally, we note that we discuss the potential consequences of the gentle beach slope on the results in lines 490-494.

In the present study, only a storm surge event induced by hurricane Ophelia was used. Is an event representative? More details should be provided in the manuscript.

** It is true that we focused on a single surge (which was not induced by hurricane Ophelia, see lines 144-146) in the observational part of the paper. In the discussion of the modelling results, lines 495-506, we point out that aeolian transport during the 5-week Aeolex-II campaign was limited during the two storm surges (October 5/6 and 29, 2017) only. During non-surge conditions we found the supply to be unlimited. Based on the modelling results, we thus infer that storm surges are the dominant conditions that result in an overprediction of measured foredune deposition volume, consistent with earlier topographic observations/suggestions in the literature. In the Discussion section we point to the need for longer-term (seasons to years) modelling to test the generality of our observions (lines 501-502). We also added that the degree of supply limitation will depend on surge characteristics, such as surge magnitude, duration and timing relative to high tide (lines 502-504).

Round 2

Reviewer 3 Report

L283-285. It is not yet clear what i and i1 mean. 

Does i indicate a single grid or a group? What do you mean "group?"

What do you mean in "For each group i of grid points with equal Fc,"

Do you mean like this?

"For each group i of sequent grids with their horizontal length equal to Fc "

L284-285: Does the following modification explains what the author means? 

This is indicated by i1-1 in Eq. (4) with i1, the first location index of the present Fc group, i.

 If so, how do you determine the first location index of the present Fc group?

Author Response

We thank the reviewer for his comments to further clarify the meaning of i and i1 and we apologize that their meaning wasn't clear from our first revision. In the following we repeat the reviewer's comments and then provide our response, starting with **. The line numbers refer to the new manuscript version with the changes highlighted in bold face. We hope that our new text clarifies the meaning of i and i1-1.

(Meaning of i)

Does i indicate a single grid or a group? What do you mean "group?"
What do you mean in "For each group i of grid points with equal Fc,"
Do you mean like this? "For each group i of sequent grids with their horizontal length equal to Fc "

** The "i" indeed refers to a group (or list) of consecutive gridpoints. The gridpoints in a group have in common that the computed value of the critical fetch (Eq. 5) is the same. We rewrote the text concerning "i" and "group" in lines 287-289: "In Eq. (4) we denote a group of consecutive grid points with the same F_c by i; in general, there will be 3 F_c groups from sea to land, namely a group with 6% < w_s <= 10% (i = 1), with 4% <= w_s < 6% (i = 2) and with w_s < 4% (i=3)." To prepare the reader for the three groups, we additionally modified a few sentences earlier in the paper, namely:
- we now define the critical fetch in line 63.
- we rewrote lines 272-276 to indicate that, in general, predicted surface moisture will decrease from its saturated value near the shoreline to its residual value on the dry beach. Given the step-function in Eq. (5) with a change in the value of "p" at a moisture content of 6 and 4%, this landward reduction in surface moisture will automatically cause the three groups as indicated in the new text. We also added here that Eq. (4) is solved at every output time step of the moisture module. Later on in the paper (line 333), we specify this output time step as 10 minutes for the present model application.

(Meaning of i1-1)

L284-285: Does the following modification explains what the author means? This is indicated by i1-1 in Eq. (4) with i1, the first location index of the present Fc group, i.
If so, how do you determine the first location index of the present Fc group?

** The reviewer correctly interpreted the meaning of i1. We have slightly rewritten his text and included it in line 291.
Because we have already computed ws at all gridpoints through Eqs. (1-3), we automatically know to which group each gridpoint belongs (see our answer to the meaning of i above). Therefore, the location i1-1 is known as well. For example, if gridpoints 80 to 110 belong to Fc group 2 (because they have a moisture content between 4 and 6%), i1-1 = 79.